# Human leishmaniasis vaccines: Use cases, target population and potential global demand

**Stefano Malvolti**[1]*, **Melissa Malhame**[1], **Carsten F. Mantel**[1], **Epke A. Le Rutte**[2], **Paul M. Kaye**[3]*

**1** MMGH Consulting, Zurich, Switzerland, **2** Department of Epidemiology and Public Health, Swiss Tropical and Public Health Institute, Basel, Switzerland; Department of Public Health, Erasmus MC, University Medical Center Rotterdam, Rotterdam, The Netherlands, **3** York Biomedical Research Institute, Hull York Medical School, University of York, Heslington, York, United Kingdom

* malvoltis@mmglobalhealth.org (SM); paul.kaye@york.ac.uk (PMK)

**Data Availability Statement:** The demand modelling has been performed in Microsoft Excel. All information and assumptions related to the calculation are included in the methodological

## Abstract

The development of vaccines against one or all forms of human leishmaniasis remains hampered by a paucity of investment, at least in part resulting from the lack of well-evidenced and agreed estimates of vaccine demand. Starting from the definition of 4 main use cases (prevention of visceral leishmaniasis, prevention of cutaneous leishmaniasis, prevention of post-kala-azar dermal leishmaniasis and treatment of post-kala-azar dermal leishmaniasis), we have estimated the size of each target population, focusing on those endemic countries where incidence levels are sufficiently high to justify decisions to adopt a vaccine. We assumed a dual vaccine delivery strategy, including a wide age-range catch-up campaign before the start of routine immunisation. Vaccine characteristics and delivery parameters reflective of a target product profile and the likely duration of the clinical development effort were considered in forecasting the demand for each of the four indications. Over a period of 10 years, this demand is forecasted to range from 300–830 million doses for a vaccine preventing visceral leishmaniasis and 557–1400 million doses for a vaccine preventing cutaneous leishmaniasis under the different scenarios we simulated. In a scenario with an effective prophylactic visceral leishmaniasis vaccine, demand for use to prevent or treat post-kala-azar dermal leishmaniasis would be more limited (over the 10 years ~160,000 doses for prevention and ~7,000 doses for treatment). Demand would rise to exceed 330,000 doses, however, in the absence of an effective vaccine for visceral leishmaniasis. Because of the sizeable demand and potential for public health impact, a single-indication prophylactic vaccine for visceral or cutaneous leishmaniasis, and even more so a cross-protective prophylactic vaccine could attract the interest of commercial developers. Continuous refinement of these first-of-their kind estimates and confirmation of country willingness and ability to pay will be paramount to inform the decisions of policy makers and developers in relation to a leishmaniasis vaccine. Positive decisions can provide a much-needed contribution towards the achievement of global leishmaniasis control.

annex. The excel file can be made available upon request by MMGH Consulting info@mmglobalhealth.org.

**Funding:** This work was funded by a Translation Award from the Wellcome Trust (Grant No. 108518; to PMK). EALR gratefully acknowledges funding of the NTD Modelling Consortium by the Bill and Melinda Gates Foundation (OPP1184344). MMGH Consulting (SM, MM, CFM) work was funded by University of York (Consulting Agreement UYPROC_582). The funders had no role in study design, data collection and analysis, decision to publish, or preparation of the manuscript with the exception of University of York (PK) contribution to the study design and preparation of the manuscript.

**Competing interests:** I have read the journal's policy and the authors of this manuscript have the following competing interests: PMK is co-author of a patent protecting the gene insert used in Leishmania candidate vaccine ChAd63-KH (Europe 10719953.1; India 315101) and is funded by the UK Medical Research Council / Department for International Development to develop a controlled human infection model for sand fly-transmitted cutaneous leishmaniasis. MMGH Consulting (SM, MM, CFM) were appointed to lead the work under contract from the University of York.

## Author summary

The leishmaniases are potentially vaccine-preventable diseases of global importance, yet no vaccines for human use have attained registration. This work sheds lights on the potential size of demand for a human vaccine for the prevention of visceral and cutaneous leishmaniasis as well as for the prevention and treatment of post-kala-azar dermal leishmaniasis. The analysis is grounded in the definition of vaccine use cases which, by transparently defining different applications of the vaccines in the immunisation programs, provides the basis for defining the target populations and vaccine characteristics. The output of this work, the first-of-its-kind for leishmaniasis, fills a critical information gap and will provide policy makers and vaccine developers with important insights into the public health relevance of a human leishmaniasis vaccine and the strengths of its commercial value proposition. Ultimately this research aims to inform future decisions on disease prioritization and on investments by key stakeholders, as well as to identify areas for further research.

## Introduction

Neglected tropical diseases (NTDs) impose a significant health and economic burden on the world's poorest populations and nations [1]. The latest Global Burden of Disease analysis for 2019 (GBD 2019) [2] estimates that the 20 NTDs recognised by the World Health Organization (WHO) account for 62.9 million disability-adjusted life years (DALYs) lost and up to 1.25 million deaths annually. A growing number of those NTDs are currently or may soon become vaccine-preventable, including Yellow Fever, Rabies, Ebola, Malaria and Dengue. Yet globally, vaccine development for NTDs progresses at a snail's pace [3], limited by scientific challenges in target antigen identification, the lack of correlates of protection and often unsuitable animal models. In addition, the lack of data on potential market size and value creation reduces financial incentives for vaccine developers to invest in those diseases [4].

In commercial vaccine development efforts, the size of the target population, the revenue potential, the required investment, the clinical development feasibility and the regulatory feasibility are the most influential drivers of decisions [5]. An in-depth understanding of the target populations and of the likely demand are critical inputs for those decisions; this is even more important in the case of NTDs that disproportionally affect low and lower-middle income countries with limited fiscal space. Before proceeding with the clinical development of vaccines for these diseases, commercial developers seek validation of the potential financial return that can be generated by taking them to market. Beyond vaccine developers, policy makers, donors and those responsible for regional health systems also need to take into account the size of the target populations and of the demand for a newly introduced vaccine. This will determine the public health impact in terms of reduction of mortality and morbidity and the financial resources required to implement the vaccination program. In low-and lower-middle income countries, those resources may be provided by the public budget or benefit from direct or indirect donor support (e.g., via Gavi, The Vaccine Alliance (Gavi)).

Among the NTDs, leishmaniasis ranks highly in terms of both mortality and morbidity. According to the GBD 2019, between 498,000 and 862,000 new cases of all forms of leishmaniasis are estimated to occur each year [2] resulting in up to 18,700 deaths and up to 1.6 million DALYs lost [2]. The leishmaniases are a group of diseases caused by infection with a protozoan parasite of the genus *Leishmania (L. Leishmania* spp *and L. Viannia* spp*)*. Visceral

leishmaniasis (VL; kala azar) is a systemic disease affecting the internal organs and is usually fatal if untreated [6]. Transmission of VL may be zoonotic (*L. infantum*) or anthroponotic (*L. donovani*). Although new treatment modalities for VL in South Asia (notably single dose liposomal amphotericin B; AmBisome) have considerably improved patient experience and outcome, with reported cure rates of up to 95% [7], treatment in other regions remains predominantly based on pentavalent antimonials [8], drugs with a number of severe limitations in convenience and outcome. Post-kala-azar dermal leishmaniasis (PKDL) is a stigmatizing disease that usually follows treatment of VL caused by *L. donovani*. PDKL patients carry a significant socio-economic and psychosocial burden. In addition, the skin of PKDL patients is a site of parasitism and data derived from xenodiagnosis [9, 10] lends support to the long held view that PKDL patients play a pivotal role in the interepidemic transmission of VL [11, 12].

Cutaneous leishmaniasis (CL) is the most common form of leishmaniasis affecting humans. This disease is considered to be a zoonosis, with the exception of *L. tropica*, which in certain areas is an anthroponotic disease. Healing of localised CL is usually achieved within 3 to 18 months without intervention, though sterile immunity is not thought to be achieved. Nevertheless, CL carries a significant burden of psychosocial risk that, once accounted for and added to the GBD estimates could lead to estimates of DALYs lost which are up to 10 times higher than current figures [13]. Chemotherapeutic options for CL have changed little in over 50 years [14]; those medicines remains expensive and questions are still unanswered about their effectiveness and safety [15]. Once cured, protection against reinfection is believed to be the norm [16, 17] and this supports the argument that vaccine-induced protection should be achievable. Localised lesions can evolve into more severe disease characterized by metastasis to mucosal sites (mucocutaneous leishmaniasis; MCL) or the occurrence of multiple (>10) discrete lesions (disseminated cutaneous leishmaniasis; DSL). Rarely, parasites may grow uncontrolled in diffuse lesions across the skin (diffuse cutaneous leishmaniasis; DCL).

The availability of vaccines against one or more forms of leishmaniasis could provide an affordable way to reduce mortality and morbidity, addressing the above-mentioned constraints. Vaccines may be deployed to prevent disease (i.e., prophylactically) or used as alternatives to or in conjunction with existing drugs (i.e., as therapeutic vaccines) for prevention of primary disease or prevention of secondary sequelae. Practical considerations including for example population at risk or alternative treatment options would dictate the relative value or prophylactic vs therapeutic vaccines. While the public health need for a vaccine exists, clinical development efforts have been limited. Numerous vaccine candidates have been evaluated in preclinical models of disease [18] but few have progressed to the clinical trial stage [19]. Currently, only one therapeutic clinical trial is ongoing (Clinicaltrials.gov NCT03969134), and a clinical-grade genetically attenuated live *L. major* vaccine is due to be manufactured in 2021 [20] to support future trials. Recent progress towards the development of a controlled human challenge model for CL [21, 22] may also provide a stimulus for the clinical development of other candidate vaccines. However, the absence of a consensus on the size of the target populations, the paucity of data to support an indication for use in each disease state and the lack of realistic demand scenarios are likely determinants of the scarce interest from the pharmaceutical industry and philanthropic donors. Estimates of the total burden of leishmaniases have been difficult due to the prevailing poor knowledge of the geographical distribution of the diseases. A further difficulty in burden estimation is the epidemic nature of the disease, leading to significant interannual variation in disease burden [23]. Those factors have made the definition of a reliable demand forecast for a leishmaniasis vaccine very challenging.

To address this critical gap, we developed a first in-depth demand forecast for a leishmaniasis vaccine that indicates that prophylactic vaccines for visceral and cutaneous leishmaniasis could have not only a solid public health value proposition but also, subject to countries

confirming their interest for the vaccine and willingness to pay, a commercial potential that can attract the interest of vaccine developers and manufacturers.

## Methods

To develop the leishmaniasis vaccine demand forecast we employed established population-based forecasting methods [24–26]. We started from the definition and estimate of the target populations for a leishmaniasis vaccine based on the use cases for such vaccine, and the evolving epidemiology of the disease. We then simulated the potential reached populations by assessing and by defining a set of assumptions on the sequence of adoption of the vaccine in endemic countries and the impact of the introduction of the vaccine on those populations. Finally, we developed the demand forecast by translating these data into doses of vaccine required, taking into account vaccine characteristics, immunisation schedule and specific programmatic issues. The logical framework of the methodology is represented in **Fig 1**.

### Use cases definition

To identify the main use cases, defined as "*the specific situation(s) in which a product or a service could potentially be used to accomplish a defined goal*", a workshop was held in May 2019 at the Wellcome Trust in London bringing together researchers involved in the development of leishmaniasis vaccines and therapies, public health professionals and other key opinion leaders. The clinical manifestations of the disease and the treatment goals, differentiated by prophylaxis (primary prevention), prevention of the development of PKDL and treatment of VL, CL or PKDL (therapeutic), were identified as key determinants of the use cases for a leishmaniasis vaccine. Consensus emerged on the factors most likely to require and / or trigger alternative approaches and distinct pathways to disease control. Vector species and subspecies as well as geography were deemed potentially relevant dimensions in future refinement of the use cases.

### Estimating the target populations

The target populations for the prevention of VL and CL were defined as the populations at risk of these diseases. For the prevention of VL (PKDL) and the treatment of PKDL, the target

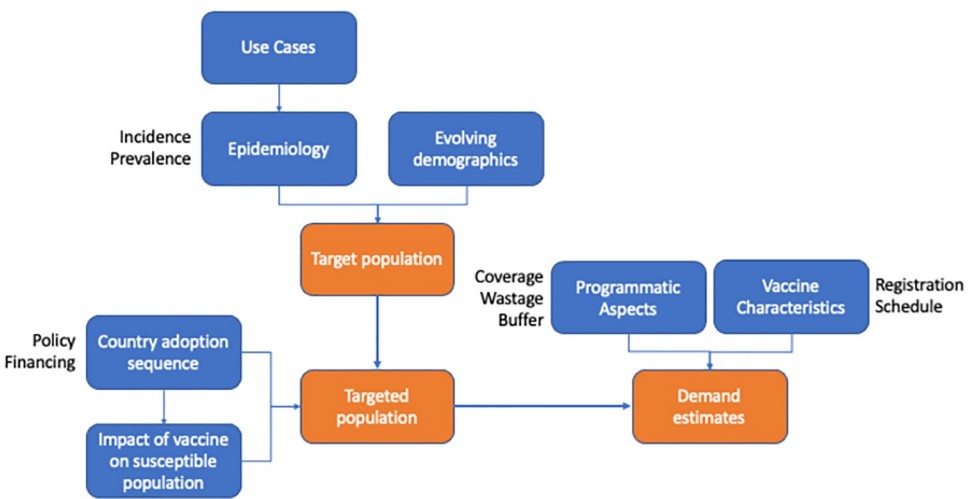

**Fig 1. Logical framework of the demand forecast methodology.**

population was defined based on VL incidence and PKDL prevalence, respectively. Data for these populations are limited and often inconsistent in terms of geographical and temporal coverage. Therefore, estimates were defined through an iterative process that started with the identification of the geographies at risk (whether entire countries or specific regions) followed by the collection of the available global, national and regional data. Estimates resulting from different sources were further scrutinized and used to define an estimate with upper and lower range limits to serve as a "base" case.

**Population at risk for VL.** 25 higher-risk countries are classified by the WHO as "high burden countries" or those with incidence above 10/10,000 according to latest estimates from the GBD of 2019 [2] (see **S1 Text**). A lower national inclusion threshold was used compared to the 1/10,000 indicated as the national target for the elimination of leishmaniasis as a public health problem, to reflect the heterogenous disease occurrence within each country and the fact that reducing incidence to the target in the risk areas will result in a subsequent reduction to even lower level of the national incidence [27]. We assumed that in those 25 countries VL burden supports the rationale for vaccine use. At-risk population data were retrieved at country level, where available, or at regional or global level [28–30]. For each of those countries, only the portion of the population considered living in at-risk areas was included as target for the vaccination program. The upper limit of the size of the at-risk population was defined based on data from 13 WHO leishmaniasis country profiles [31] (data from 2014, 15 and 16 depending on the country) integrated with data from Pigott et al [28] that includes all geographical areas (at national or sub-national level) which could become endemic in the future. Additional validations were performed for the total population of the Indian subcontinent (India, Bangladesh and Nepal) [32]. The consolidated estimate for the 25 countries was allocated to the year 2014. The base case was centred around the latest available global WHO estimates of the overall population at risk of all leishmaniases (for 2010) under the assumption that 100% of that population is at risk of VL. The assumption is consistent with the analysis by Pigott et al [28] where the difference in size of VL and CL at-risk populations was minimal. The lower limit of the range was based on the 2009 global estimates developed by DNDi [30]. Since the year of reference was different for the different data sources, projections up to 2018 were built for each dataset, applying a population growth rate of 1.80%, corresponding to the 2010–2015 regional growths as reported by UN/DESA [33] and weighted based on regional VL cases from the GBD estimates 2019 for Asia inclusive of the Middle East, Africa and Latin America.

**VL incidence.** Information for the 9 VL endemic countries relevant for PKDL was retrieved from the latest GBD 2019 [2] for the period 2010–2019. Other estimates of new VL cases were sourced from Alvar (personal communication) for India and East Africa for 2018 and from Mondal et al. [34] for India for the period 2005–2008.

**PKDL prevalence.** Estimates of the rate of occurrence of sequelae of VL cases were used to define PKDL incidence. Different regional estimates were used for Asia [35] and Africa with high, low and medium points being defined. From the PKDL incidence, the 2016 PKDL estimated prevalence was derived considering an average 3.5 years duration of the lesions as indicated by Mondal [34].

**Population at risk for CL.** 34 high-risk countries were selected, either those identified by WHO as high burden countries or those with an incidence above 5/100,000 according to the latest GBD estimates for 2019 [2]. For India, only 2 states with high endemicity (Rajasthan [36] and Kerala [37]) were included. Similar to VL, these countries and states were assumed to be those where CL burden supports the rationale for vaccine introduction. For each of those countries, only the portion of the population living in at-risk areas was included as a target for the vaccination program. The upper limit of the size of the at-risk population was defined

based on data from 12 WHO leishmaniasis country profiles [31] (data from 2014, 15 and 16 depending on the country) and the consolidated regional 2014 estimate for the 12 countries in the American region [38], integrated with data from Pigott et al. [28] from 2014. The consolidated estimate for the 34 countries was centred on the year 2014. The mid-point of the range was based on the global estimates from DNDi of the total cases of leishmaniasis [39], under the assumption that the countries included in the estimate were all at risk for CL. The lower limit of the range was based on the global WHO estimates for 2010 [29] of the overall population at risk of all leishmaniases under the assumption that 100% of these populations are at risk of CL. Similar to VL, projections up to 2018 were built applying a 1.67% population growth rate for the period 2010–2015. This rate corresponds to the regional estimates as reported by UN/ DESA[33] weighted based on the latest available regional CL cases from GBD 2019 for Asia inclusive of Middle-East, Africa and Latin America.

The four estimates above were then projected through the year 2040 by using the weighted regional population growth rate (based on regional shares of the respective disease for the period 2019–2040 as estimated by the UN DESA[33]. Specifically, a growth rate of 1.44% was used for the population at risk of VL and 1.27% for the population at risk of CL. For the projection of the VL cases, a wave-shaped incidence curve was applied for the period 2017–2040 reflecting an epidemiological trend with 5 years of decline followed by 5 years of growth, but less accentuated than the cyclical incidence variations induced by El Niño [40–42], and using the UN DESA regional average annual growth rates for the period 2010–2016 (with a positive sign in the growth and a negative sign in the decline years).

## Modelling the targeted population

**Adoption sequence in the targeted countries.** The adoption curve indicates the growing proportion of the targeted population that can be reached with the vaccine as result of country introduction decisions. Country adoption decisions are often guided by an assessment of the benefits of vaccination under different conditions by the National Immunisation Technical Advisory Group (NITAG) based on the country and region-specific context. The likelihood of NITAG recommendations for vaccine introduction in different countries was accounted for by including only high-risk countries based on incidence thresholds as indicated earlier.

In contrast to prophylactic vaccines, preventative and therapeutic vaccines do not have the same requirement for centralized decision-making. The countries where PKDL is endemic are thus assumed as introducing the vaccine.

In view of the early state of development, interest of individual countries for introduction of a leishmaniasis vaccines cannot easily be assessed, hence a proxy was identified to define the sequence of adoption in the selected countries. Since 21 of the 25 countries (84%) which are likely targets for the introduction of a VL vaccine and 13 of the 34 countries (38%) which are likely targets for a CL vaccine are or were eligible for Gavi support, the sequence of introduction in the Gavi countries for Pneumococcal Conjugate Vaccine [43] was selected as the most meaningful proxy.

**Impact of vaccine introduction on target populations.** The introduction of a leishmaniasis vaccine that triggers an effective host protective immune response will decrease the size of the population at risk. We assumed that the introduction of a VL prophylactic vaccine would decrease populations at risk and also reduce transmission in areas with exclusive anthroponotic transmission (South Sudan, Sudan, Somalia, India, Nepal, Bangladesh and the Xinjiang region in China). In those countries, it was assumed that the decline of the population at risk would take place linearly and that immunisation would progressively be stopped once all blocks or districts have reached a VL elimination target of less than 1 VL case per 10,000 per

year. This is estimated starting from the sixth year after vaccine introduction, spreading over a period 4 years. In areas where VL transmission is zoonotic, vaccination is assumed to continue unchanged. In a similar fashion, we assumed that a CL prophylactic vaccine, while protecting individuals, will have less impact on transmission given the zoonotic nature of this disease, hence immunization will not be stopped in any CL at-risk area. Finally, the impact of a VL prophylactic vaccine on VL incidence and thus, indirectly, also on the size of the target population for the prevention of PKDL was estimated referring to recent disease transmission modelling work for India, where anthroponotic transmission was assumed [44]. In that model (under the assumption that asymptomatic individuals also contribute to VL transmission), the reduction in the number of cases following prophylactic vaccination with a 50% efficacious vaccine covering 100% of the population was estimated to be 30%, 50% and 62% in years 1, 2 and 3 following the introduction of the vaccine [44].

The impact of a PKDL preventative vaccine on PKDL incidence was modeled taking into account the following factors: i) the year of introduction of the vaccine, ii) the proportion of the total population reached as a result of a vaccine deployment spread out over 2 years, and iii) the estimated reduction of VL cases. As a result, a 50% reduction in year 2 and a 90% reduction from year 3 onward is used to estimate the reduction in the number of cases in each of the 3 scenarios with year 1 being the year of introduction of the vaccine.

For the impact of a PKDL therapeutic vaccine on PKDL incidence, no changes to the target population were assumed.

## Modelling the demand for a leishmaniasis vaccine

With the target population estimates established and expected changes to the size of those populations modelled, we subsequently defined a number of assumptions related to the vaccination program and to the vaccine characteristics.

**Date of registration of the vaccine.** Based on the current status of vaccines in clinical development, we took the earliest year of first registration for both VL and CL preventative vaccines as 2029. We then assumed 12 months for obtaining WHO prequalification (PQ) to allow procurement through the United Nations system. The PQ date is based on the assumption that the vaccine will be developed by an experienced manufacturer and is consistent with assumptions used by the WHO in similar analyses [45]. Thus, 2030 was assumed as the year of start of a phased global roll-out for both indications. A PKDL therapeutic vaccine (ChAd63-KH) is in Phase IIb therapeutic trial in in Sudan [46] while Phase II trials of the same vaccine to prevent PKDL in previously treated VL patients may begin in 2021. Should one or both of these approaches prove successful, such a vaccine might progress through other trials to allow registration in 2027.

**Vaccine characteristics and immunisation schedule.** For estimating the demand, key vaccine product characteristics were assumed. Target product profile (TPP) assumptions relevant to the calculation of a demand forecast are shown in **Table 1**.

Of special importance is the assumption of cross-protection of a vaccine against visceral and cutaneous leishmaniasis, as suggested by studies with different *Leishmania* species in experimental models [47, 48]. Assumptions regarding the duration of protection and number of vaccination series (the number of times a person undergoes one full vaccination cycle) were made based on historic evidence, regulatory assumptions and practical considerations. A minimum of five years duration of protection was assumed based on the consideration that a shorter duration would likely render the vaccine too cumbersome for implementation and too expensive for use in view of the number of doses required to maintain protection for the first 15 years of life (e.g., the age groups with the highest disease burden). In addition, key

**Table 1. Selected TPP assumptions relevant for the demand forecast.**

| Indication | Age for first dose | Duration of Protection | Nr. Doses per Series | Series (years of age) |
|---|---|---|---|---|
| VL prophylactic | 1 year | 5 years | 2 | 3 (1, 6 and 11) |
| CL prophylactic | 1 year | 5 years | 2 | 3 (1, 6 and 11) |
| CL/VL catch-up | | | 2 | 1 |
| PKDL therapeutic | NA | Lifelong | 1 | 1 |
| PKDL preventive | NA | Lifelong | 1 | 1 |

assumptions were made regarding the immunisation schedule, including broad age-range catch-up campaigns at start of the vaccine roll-out in each country with the goal of reducing susceptible unprotected populations in an efficient and timely manner. To capture a main area of uncertainty, a second more conservative scenario has been run simulating a first priming dose followed by 1 booster dose after 5 years and a second booster after 10 years for a total of 3 doses (instead of the 6 of the base case).

**Vaccine coverage.** The maximum achievable coverage was estimated for different vaccine schedules based on coverage achieved in routine vaccination in the same age groups [49, 50] or coverage achieved in campaigns in the relevant regions [51] as illustrated in **Table 2**.

As a simplifying assumption, the time required to achieve the full coverage within each country was set at 12 months.

A detailed review of the calculations is available in **S1 Text**.

## Results

### Determination of use cases

Eleven use cases emerged as a result of the review (**Table 3**).

Three of these use cases were deprioritised in view of their limited size and impact on the total demand. For each of the remaining eight use cases, the most appropriate delivery strategy was defined according to the target geography and goal (prophylaxis, prevention or treatment). Due to the focal distribution of leishmaniasis and the high population mobility in most of the endemic areas, delivery strategies were assumed to focus on at-risk populations rather than at-risk areas. In finalising the use cases, the following two aspects were also considered: (a) a CL therapeutic vaccine will target a subset of the population also targeted by the CL prophylactic vaccine (the population at risk of CL); (b) vaccine development efforts will most likely not focus on geographically limited indications. As result of those considerations, the eight prioritised use cases were further consolidated into four by eliminating the differentiation by geography and co-morbidity for the CL use cases (**Table 4**).

These four use cases and the related delivery strategies provide the foundation for the assessment of the size of the demand for leishmaniasis vaccines given the different indications.

**Table 2. Delivery strategy and related coverage assumptions (source: WHO/UNICEF coverage estimates for 2018, WHO Measles campaign overview 2000–2020).**

| Indication | Delivery | Coverage |
|---|---|---|
| VL & CL prophylactic | Routine at 9 months-2nd year of life | 73% |
| | Routine at 6 years– 11 years | 68% |
| | Routine for adults | 45% |
| | Campaign delivery (including catch-up) | 90% |
| PKDL therapeutic & preventive | Following delivery of VL or PKDL treatment | 93% |

**Table 3. Leishmaniasis vaccine use cases definition and prioritisation.**

| Clinical present. | Species / geographies | Goal | Use case | Rationale |
|---|---|---|---|---|
| VL | *L. donovani/infantum* All geographies | Prophylactic | VL prophylaxis | *Broader impact* |
|  |  | Therapeutic for HIV+ | 2nd line treatment of VL in HIV+ | *Deprioritized in view of the difficulty of treatment* |
| PKDL | South Asia and East Africa | Preventive | Prevention of VL relapse (PKDL) | *Seriousness of the diseases* |
|  |  | Therapeutic | 1st line treatment of PKDL |  |
| CL | *L. tropica/aethiopica* East Africa and Middle East* | Prophylactic | CL prophylaxis | *Poor available treatment, uncontrolled* |
|  |  | Therapeutic | 1st line treatment of CL |  |
|  | *L. (Viannia) braziliensis* Brazil / Americas | Prophylactic | CL prophylaxis | *Morbidity can be addressed through treatment* |
|  |  | Therapeutic | 1st line treatment of CL |  |
|  | Other species | Prophylactic | CL prophylaxis | *Deprioritised in view of the limited size of the target population* |
|  |  | Therapeutic | 1st line treatment of CL |  |
| MCL-DSL-DCL | All species | Therapeutic | 1st line treatment of MCL-DSL-DCL | *Deprioritised in view of the limited size of the target population* |

## Target population estimates

Using a variety of source data (see Methods), we generated upper and lower limits of target populations that are at risk of developing VL or CL and who might therefore benefit from a prophylactic vaccine. Similarly, upper and lower limits are estimated for preventative and therapeutic vaccines for PKDL. In addition, we identified a base case or central-point estimate of the population at risk (**Table 5**).

For both VL and CL prevention, the 2018 estimates of populations at risk ranged from 647 million to 235 million for VL and from 1 billion to 399 million for CL. Based on the assumption of cross-protectivity and taking into account the population of the 4 countries (Brazil, India, Paraguay and Sudan) included in the analysis that are endemic for both manifestation of the disease, approximately 29% of the population at risk of CL is considered also at risk of VL and was therefore deducted from the calculation of the CL target population. Estimates for the prevention of PKDL ranged from 31,892 to 12,635 and for treatment of PKDL from 6,141 to 2,460, emphasizing a marked difference in scale for these different indications.

## Potential demand estimates for VL and CL prophylactic vaccines

Based on the above assumptions, we calculate that peak combined potential demand for VL and CL would be approximately 190 million doses in 2033 (of which approximately 1/3 are for

**Table 4. Final set of use cases and delivery strategies.**

| Use case | Species / geographies | Delivery strategy/ies |
|---|---|---|
| 1. Prevention of VL | All geographies | Routine or campaign delivery in target at-risk populations (identified via endemicity mapping) Outbreak response via ring vaccination |
| 2. Prevention of CL | All geographies | Routine or campaign delivery in target at-risk populations (identified via endemicity mapping) |
| 3. Prevention of PKDL | All geographies | Adjunct(following) VL treatment |
| 4. Treatment of PKDL |  | Same as current PKDL treatments |

**Table 5. Target population (*30% of this population is assumed being also at risk of VL).**

| Indication | Target Population | Range | Sources | 2018 |
|---|---|---|---|---|
| **Prevention of VL** | Population at risk of VL | Upper limit | WHO country estimates for 2014–15 and Pigott 2014 | 647,000,000 |
| | | Mid-point (base case) | WHO TRS 2010 | 404,000,000 |
| | | Lower limit | DNDi 2009 | 235,000,000 |
| **Prevention of CL** | Population at risk of CL* | Upper limit | DNDi 2018 | 1,000,000,000 |
| | | Mid-point (base case) | WHO country estimates for 2014–15 and Pigott 2014 | 773,000,000 |
| | | Lower limit | WHO TRS 2010 | 399,000,000 |
| **Prevention of PKDL** | VL cases | Upper limit | GBD, 2017, WER 2018, Alavar 2012 | 31,892 |
| | | Mid-point (base case) | | 20,730 |
| | | Lower limit | | 12,635 |
| **Treatment of PKDL** | PKDL cases | Upper limit | WER 2018, Alavar 2012 with disease specific assumptions from Zijstra 2016, Kaye 2019, Mondel 2018 | 6,141 |
| | | Mid-point (base case) | | 4,006 |
| | | Lower limit | | 2,460 |

VL) (**Fig 2**). Demand shows a variable profile as a result of the impact of the large catch-up campaigns aimed at multiple age cohorts (10 cohorts for VL and 25 cohorts for CL): as the population covered by the initial catch-up campaigns grows older, the portion of the 0–15 age group requiring boosting via routine immunization will grow. After 2034, the increasing impact of the vaccine will result in the progressive reduction in size of the VL at-risk population offsetting the additional populations introducing the vaccine. As a consequence, demand is predicted to decline progressively until 2038, reflecting the elimination [52] of VL in the

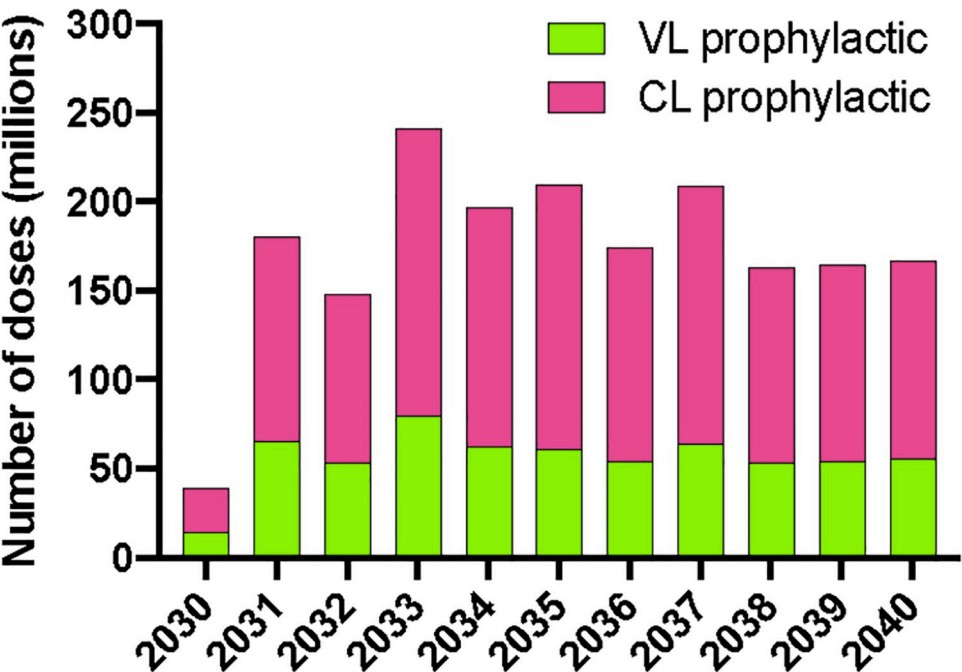

**Fig 2. Vaccine demand–total number of yearly doses required at global level for a vaccine with VL and CL indications (details of the methodology provided in S1 Text).**

Indian subcontinent, and thereafter to stabilise at around 160 million doses per year across the two indications. Such an assumption, based on the historical trend of the last decade requires continued investment in surveillance and introduction of better diagnostic methods[53].

Estimates in single years are a function of many variables and may have limited predictive value and the portion of demand that will be served would most likely be capped at some level due to limits of manufacturing capacity. The cumulative number of doses over the 11 years between 2030 and 2040 is thought to provide a reasonable and relevant estimate resulting in a demand for approximately 1.6 billion doses across the 2 disease states (approximately 500 million for VL and 1.1 billion for CL) or an average of 145 million doses per year. It is important to note that this estimate is significantly impacted by the assumption that six doses will be required for each person vaccinated. In the more conservative scenario where only 3 doses are required, those numbers are reduced by half (800 million doses over the 11 years across the 2 disease states).

Of interest is the split between campaigns (for catch-up vaccination at start) and routine use, with the former being more volatile and dependent on the number of cohorts targeted and the timing of catch-up campaigns in different countries. It is estimated that 39% of the doses (approximately 615 million doses) will be used in campaigns to reach 5–14 year olds for protection against VL and 5–29 year olds for protection against CL in view of the different age distribution of the disease [54] while the remaining 61% (approximately 980 million doses) will be used in routine immunisation in areas with high CL and VL prevalence (because of the larger and stable population reached) (**Fig 3**).

Two additional simulations were also performed. One simulation used the lower population scenario as a way of reflecting a conservative block-focused immunization strategy and a significant negative impact of vaccine hesitancy. A second simulation used the higher population scenario so as to capture a more extensive definition of risk-areas and also incorporating a broader impact of climate change on the spread of the disease. The first scenario was reflected in a reduction of the total doses over the 11 years from 1.6 billion to 858 million (301 million for VL and 557 million for CL); the second scenario resulted instead in an increase to 2.2 billion doses over the 11 years (830 million for VL and 1.4 billion for CL).

### Potential demand estimates for PKDL targeted vaccines

Assuming implementation of an effective VL prophylactic vaccine, total PKDL demand will peak at 27,300 doses in 2028 with approximately 90% of demand for preventive use (**Fig 4**). For therapeutic use, demand is estimated at 3300 doses in 2027 (at the time of vaccine introduction) declining to less than one hundred doses by 2036. Overall demand will further decline to less than 1,000 doses in 2039 as result of the cumulative impact of preventative vaccine use, compounded by the reduction in VL cases driven by the introduction of the VL prophylactic indication. The total demand for a PKDL therapeutic vaccine over the 13 years will be 7,200 doses, while the total demand for a vaccine to prevent PKDL over the same period will be 161,000 doses. Of interest is the scenario where a stand-alone therapeutic / preventative vaccine for PKDL is developed in the absence of an effective prophylactic vaccine for VL and hence with no vaccine-related reduction in VL cases. Under those circumstances, the total potential demand for a PKDL vaccine doubles to more than 330,000 doses over the period.

### Discussion

In order to fully ascertain the potential interest of developers for a program aimed at developing a vaccine against leishmaniasis, we conducted an in-depth analysis of the potential global demand for leishmaniasis vaccines that is generally agnostic to the nature of the vaccine.

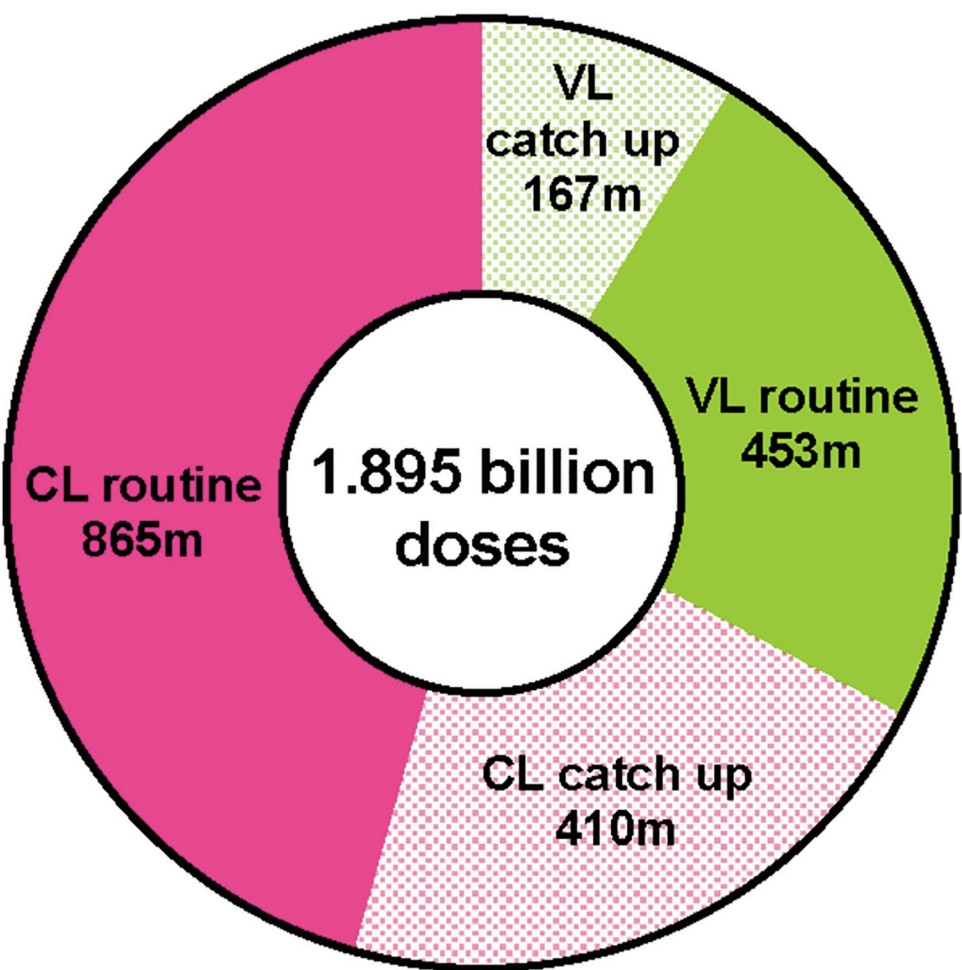

**Fig 3. Vaccine demand: total number of doses for the period 2025–2040 stratified by delivery strategy (routine immunisation and catch-up campaign at start) and by indication (VL and CL).**

The assessment of the use cases for leishmaniasis vaccines emphasizes four main indications: two for the prevention of VL and CL respectively, and two for the prevention and treatment of PKDL. In endemic countries, those vaccines will reduce cases of disease and decrease community transmission. Based on the estimates of the populations at risk and the consequent demand, prophylactic VL and CL vaccine indications are the most attractive scenarios both from a public health and commercial standpoint. A vaccine that is cross-protective against VL and CL has the strongest value proposition, whilst only one indication still allows for a significant demand size and public health impact. PKDL indications also retain their importance from a public health standpoint or in view of existing progress in clinical development that may lead to the earlier availability of this vaccine.

Availability of a vaccine for one or more of these indications will provide an opportunity to reduce the threat of leishmaniasis as a public health problem in 57 countries (split as follow: India and Sudan, for the 3 manifestations of the disease, 4 countries for VL and CL, 7 countries for VL and PKDL, 14 countries for VL only, and 30 countries for CL only). This includes 22 countries in the African continent, 14 in the Americas, 19 in Asia, and 2 in the European region. At the same time 3 of these countries are High-Income (HICs), 18 Upper Middle-income (UMICs), 19 Lower-Middle-Income (LMICs) and 15 Low-income (LICs) [55] as

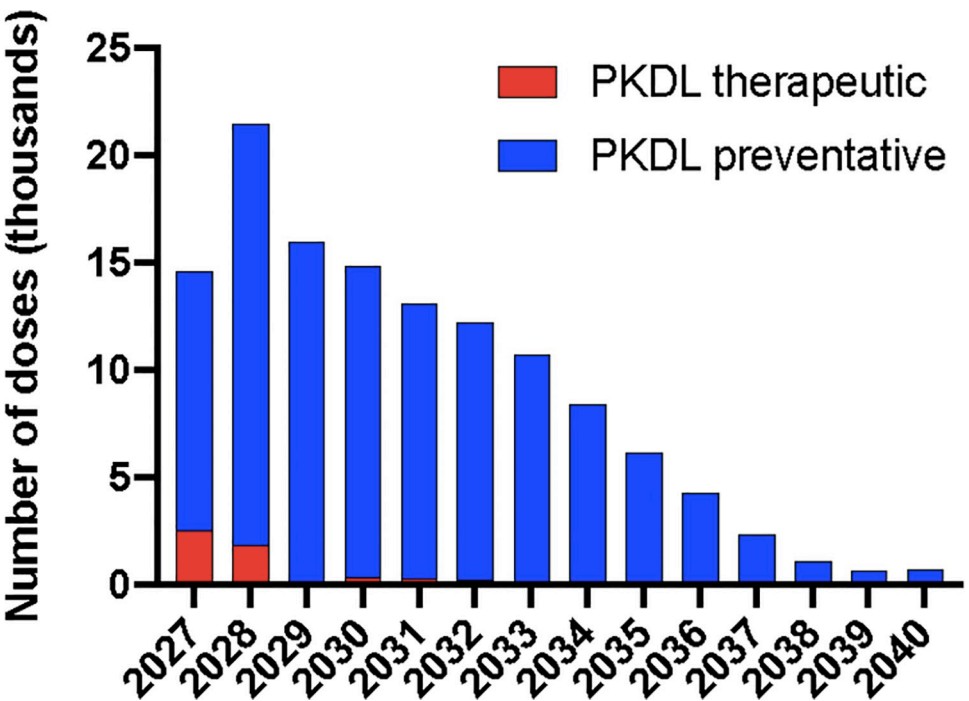

**Fig 4. Vaccine demand–total number of yearly doses required at global level for a vaccine with PKDL therapeutic and preventative indications (in the scenario of a vaccine with VL indication also being available).**

defined by the World Bank income classification. Of the latter two groups, 22 countries are, and 10 have previously been, eligible for Gavi support including funding for vaccine procurement. It is assumed that HICs or UMICs would potentially be able to self-finance, while LMICs and LICs could benefit from the financial support of Gavi and other donors.

The total population potentially targeted by leishmaniasis vaccines approximates 1.1 billion in 2030 by the time of introduction of the vaccine. On the basis of data from 2018, the population at risk of VL (404 million with a range between 235 and 646 million) and at risk of CL (765 million with a range between 392 million and 1 billion) represents the near totality of the potential target for the vaccine while the numbers of VL cases in the targeted countries (16,129 new cases per year for 2030 with a range between 2,843 and 25,068 cases) and of PKDL (3,104 new cases per year for 2030 in absence of a vaccine, with a range between 600 and 4,752 cases) are much more limited. It is evident that vaccine use in such large populations could result in significant public health benefits.

The vaccine supply required over 10 years (2030–2040) for the implementation of immunisation programs against VL or CL could be substantial: approximately 1.1 billion doses of vaccine against CL (range: 0.5–1.4 billion doses) and 0.5 billion doses of vaccine against VL (range: ~ 0.3–0.8 billion doses). Markets of this size have the potential to attract the interest of vaccine manufacturers, even more so in view of the mix of economies involved, some of which may self-finance a respective immunization program. For this interest to materialise, utmost transparency will be necessary on the countries' willingness and ability to pay for the vaccines, as well as clarity on WHO recommendations for vaccine use. Establishment of incentive mechanisms, such as Advanced Market Commitments [56] and Priority Review Vouchers [57], can also play an important incentivizing role. This will be critical for supporting the progress of current vaccine development efforts.

It is not yet established whether a vaccine that prevents clinical VL will also prevent the development of PKDL. If this were not the case, making available a vaccine that is selective for PKDL may be more challenging in view of the more limited potential demand. However, a different approach may be pursued for this indication leveraging the limited demand as an opportunity instead of a constraint. Regulatory frameworks for orphan or compassionate use indications could be exploited and combined with limited manufacturing needs and investments, reducing greatly the cost and the risk for a manufacturer. Such a focused approach to vaccine financing should be explored to address the burden of PKDL.

To support countries in their assessment of the public health impact of a leishmaniasis vaccine and prospective manufacturers of the commercial value proposition of those vaccines, clarity on target populations and demand is paramount. The Hib Initiative, as well as the Pneumococcal and Rotavirus Vaccine Accelerated Development and Introduction Plans (PneumoADIP and RotaADIP) provide examples on how a concurrent effort on data generation, clinical development and public advocacy was able to raise awareness on diseases with a heavy burden on populations of LMICs and LICs [58]. Such an approach is even more critical for neglected tropical diseases such as leishmaniasis.

The limitations of this analysis, as for any forecasting exercise, relate to various elements of uncertainty. Firstly, the forecast is dependent on the estimates of the at-risk populations in endemic or potentially endemic areas as well as on the assumed future incidence and prevalence of the disease. With current leishmaniasis control programmes in place in certain parts of the world, such incidence will depend on the success of these programmes. There is no clear consensus among experts and estimates show a high degree of variability while an agreed definition of the target population at risk, by countries or sub populations of countries, is missing. Secondly, assumptions concerning the probability of country vaccine introduction are subject to a high degree of uncertainty given that vaccines will likely not be available for another decade. Thirdly, the estimate of the impact of the use of vaccines in the target populations is based on a set of assumptions while awaiting the outcomes of vaccine trials. Work done by Erasmus MC (University Medical Center, Rotterdam) provides information on the potential impact of various VL vaccines on VL incidence in an Indian setting [44], but not elsewhere. Whilst our modelling assumptions include transmission by asymptomatic cases of VL, a recent study from India suggests that this may be more limited than previously thought [10]. Studies of transmission competence across the disease spectrum in other regions where VL is endemic are clearly warranted. Additional simulations of the impact of vaccines in CL and VL populations at risk (including the likelihood of interruption of transmission and of the impact of a VL prophylactic vaccine), will further improve our understanding of population dynamics and vaccine effectiveness. Finally, a number of initial assumptions were made on variables such as the sequence of country introduction, vaccine uptake, achievable and desired coverage and vaccination schedules. Changes in these parameters and the priority countries introducing the vaccines will impact the forecasted demand. As an example, the impact of a different immunisation schedule, as captured in the conservative scenario described in the results section (with a reduced series of 3 doses instead of 6), has the potential to reduce demand by up to 50%. Getting closer to registration, more precise information about vaccine product characteristics, and the likely program designs and country interest will allow for more refined estimates.

In conclusion, there is a growing consensus on the need for a vaccine against leishmaniasis to achieve a reduction of the burden of disease [59–61]. Clarity about the prospective size of the vaccine demand in terms of target population and number of doses required is crucial to inform decisions of manufacturers, donors and countries. Our first-of-a-kind analysis provides a global estimate of the potential demand for leishmaniasis vaccines across a set of

different indications. Subject to prioritization of country and global decision makers, a leishmaniasis vaccine with a VL and/or CL indication could not only provide a significant contribution to the reduction of the burden of NTDs but also has the potential of being an interesting commercial prospect for vaccine developers. Further analyses to confirm the likelihood and strength of interest of country decision makers in prioritising leishmaniasis vaccines in their adoption decisions are warranted. This will enable further clarification on the potential reduction in the burden of the disease and cost-effectiveness of these vaccines.

## Supporting information

**S1 Text. Supplementary text–methodological annex.**
(DOCX)

## Acknowledgments

The authors of this article wish to thank the members of the expert group who provided insights and guidance in the development of the use cases and who critically reviewed the manuscript: Jorge Alvar (DNDi), Simon Croft (LSHTM), Tom Evans (Vaccitec), Nirmal Kumar Ganguly (ICMR), Birgitte Giersing (WHO), Bethan Hughes (Wellcome Trust), Christiane Juhls (Mologen), Paul Kaye (Univ. of York), Ahmed Musa (Sudan MOH), Richard Muscat (Wellcome Trust), Steve Reed (IDRI). In particular, we thank Jorge Alvar, Simon Croft and Nirmal Kumar Ganguly who dedicated time to validate some of our assumptions. Finally, we are grateful to the Wellcome Trust for supporting expert meetings during the project timeline and allowing for additional fruitful interactions.

## Author Contributions

**Conceptualization:** Stefano Malvolti, Melissa Malhame, Paul M. Kaye.

**Data curation:** Stefano Malvolti, Melissa Malhame.

**Formal analysis:** Stefano Malvolti, Melissa Malhame.

**Funding acquisition:** Stefano Malvolti.

**Investigation:** Stefano Malvolti, Melissa Malhame.

**Methodology:** Stefano Malvolti, Melissa Malhame, Carsten F. Mantel, Paul M. Kaye.

**Project administration:** Stefano Malvolti.

**Resources:** Stefano Malvolti, Melissa Malhame.

**Software:** Stefano Malvolti, Melissa Malhame.

**Supervision:** Stefano Malvolti, Paul M. Kaye.

**Validation:** Stefano Malvolti, Melissa Malhame, Carsten F. Mantel, Epke A. Le Rutte, Paul M. Kaye.

**Visualization:** Stefano Malvolti, Melissa Malhame, Paul M. Kaye.

**Writing – original draft:** Stefano Malvolti, Paul M. Kaye.

**Writing – review & editing:** Stefano Malvolti, Melissa Malhame, Carsten F. Mantel, Epke A. Le Rutte, Paul M. Kaye.

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
