## [Decision Letter · Decision Letter 0]

15 Jun 2021

Dear Mr. Malvolti,

Thank you very much for submitting your manuscript "Human leishmaniasis vaccines: use cases, target population and potential global demand" for consideration at PLOS Neglected Tropical Diseases. As with all papers reviewed by the journal, your manuscript was reviewed by members of the editorial board and by several independent reviewers. The reviewers appreciated the attention to an important topic. Based on the reviews, we are likely to accept this manuscript for publication, providing that you modify the manuscript according to the review recommendations. 

Sincerely,

Dhafer Laouini

Associate Editor

Eric Dumonteil

Deputy Editor

Reviewer's Responses to Questions

**Key Review Criteria Required for Acceptance?**

**Methods**

-Are the objectives of the study clearly articulated with a clear testable hypothesis stated?

-Is the study design appropriate to address the stated objectives?

-Is the population clearly described and appropriate for the hypothesis being tested?

-Is the sample size sufficient to ensure adequate power to address the hypothesis being tested?

-Were correct statistical analysis used to support conclusions?

-Are there concerns about ethical or regulatory requirements being met?

Reviewer #1: See below General Comments. I may have a different opinion of the population at risk for VL and this is reflected my general comments. I am not suggesting however the authors change methodology used, but to consider my interpretations of the populations at risk for VL

Reviewer #2: Yes

Reviewer #3: The methodology could have been more clear. No models are proposed. Only speculative numbers are given. The data used for the prevalence and burden of the diseases (VL/CL) are old and after free diagnostics and treatment in many countries like in Indian subcontinent, these numbers may be too optimistic for projecting the market for vaccine.

**Results**

-Does the analysis presented match the analysis plan?

-Are the results clearly and completely presented?

-Are the figures (Tables, Images) of sufficient quality for clarity?

Reviewer #1: Overall the results are clearly presented.

Reviewer #2: Yes

Reviewer #3: Probably a more clear analysis with current strategies for kala-azar control in Indian subcontinent and overlapping prevalence of VL an CL need to be dissected out more clearly. Moreover, in CL cases, there will be vaccine hesitancy as the disease is neither fatal nor considered so disfiguring in rural males and many of these cases are even reluctant to take treatment. 

I think a section on limitation of the models and background data used should have been better.

**Conclusions**

-Are the conclusions supported by the data presented?

-Are the limitations of analysis clearly described?

-Do the authors discuss how these data can be helpful to advance our understanding of the topic under study?

-Is public health relevance addressed?

Reviewer #1: See below general comments

Reviewer #2: yes

Reviewer #3: In my opinion it is too optimistic estimation for vaccine developers. This should be used with caution.

**Editorial and Data Presentation Modifications?**

Reviewer #1: (No Response)

Reviewer #2: (No Response)

Reviewer #3: (No Response)

**Summary and General Comments**

Reviewer #1: The paper from Malvolti et al is a timely paper on the need to pursue vaccine development of the neglected tropical diseases of which leishmaniasis represents a prominent member. A major contribution of this paper is to provide much needed information on the target population and global demand. This will impower vaccine producers, policy makers and funders with the knowledge to make informed decisions about the prioritization of vaccines for leishmaniasis and post kala azar dermal leishmaniasis (PKDL). Overall this is a well written manuscript that will generate considerable interest and debate in the field. 

I do not doubt the calculations used to generate the number of vaccine doses arrived at in Figures 2 and 3. This is higher that I had expected. Considering line 188, I wonder if the authors took into account that in countries such as India with cases greater that 1/10000, this is at the block level, not the country level. Therefore, vaccines could be provided in districts with endemic blocks, not the entire country. There is in fact relatively few blocks in Bihar state with >1/10,000 . For example, the Bihar state has the highest number of cases in India. Bihar has about 30 districts and each district has about 100-200 blocks. One vaccine strategy would be to vaccinate only those districts with endemic blocks having >1/10,00 cases, not the entire state. If the focus was at the district/block level, how would this scenario impact the numbers arrived at in Figures 2-3? Is this something the authors need consideration?

Minor comments

I suggest the authors justify why a therapeutic vaccine for PKDL but not VL. For example, there are already good treatments for VL but not for PKDL. 

Does COVID vaccine development and implementation impact on the future NTD vaccine landscape? Are funders/policy makers now more aware of the importance of vaccines? 

Lines 93-95; Provide a reference for these number of cases and DALY's lost?

Lines 98-99; there is no evidence that VL; L. donovani infection is zoonotic; only L. infantum

Line 112-113 and lines 118-119 appear to be contradictory. Please clarify, is there immunity against re-infection for CL?

Line 121: MCL is uncommon following CL, should be more specific about Leishmania species associated with MCL

Reviewer #2: Leishmaniasis is one of the top neglected tropical diseases that impacts large number of people in different parts of the world. Some of the manifestations of this disease are fatal if not treated. However, over the years the drugs that have been developed have not been able to control the disease because of the drug toxicities and development of parasite drug resistance due to lack of compliance. In addition, attempts have been made to control insect vector (sand fly) through spraying the insecticides in the affected areas. These efforts have not been sustained and hence the resurgence of cases over the areas. There is also the lack of control of reservoirs for some form this disease. Therefore, as with other infectious diseases, vaccines could control the disease in a long run. However, Leishmaniasis is the disease of poor and vaccine manufacturers do not see an impetus to develop such vaccines for the lack of higher returns for their investment. 

In this timely well written article, the authors have highlighted the need for pan Leishmania vaccine which can have a significant impact on the control of the disease and could achieve the elimination goal for visceral form of the disease as set forth by WHO. They have developed a model to show the global demand for the vaccine in terms of doses needed for control, over a period of certain time frame, of three main form of the Leishmania disease which includes visceral leishmaniasis (VL), cutaneous leishmaniasis and post kalazar dermal leishmaniasis (manifestation of treatment failure of VL). They also have calculated the needed for vaccine doses to sustain the control of this disease. Importantly the input information in developing model was based on the discussion among the experts in the Leishmania field, clinicians, and the policy makers. The authors have highlighted a few potential vaccine candidates which are being pursued at the preclinical and clinical studies by various investigators. The assumptions in the input data for such modelling to predict the size of the vaccine demand is appropriate and hopefully will generate interest by the vaccine manufacturers to develop a pan Leishmania vaccine. 

Following are the comments:

1. The authors need to pay attention while discussing the size of the population at risk for CL only to focus on the people living in at risk areas. This could be underestimation since people move from adjacent nonendemic areas into the endemic areas due to many reasons. 

2. The assumption of less impact of CL vaccine because of the zoonotic nature of transmission may be valid, however similar assumption can be made for VL in the endemic areas of the world where it has been shown that there is zoonotic transmission especially in Africa. 

3. Table 1. Have the authors thought of therapeutic use of VL or CL vaccine for other co-infections in the endemic areas besides HIV? 

4. Table 2. While discussing use cases and targeting population for vaccination, it may be important to consider broader catchment areas for vaccination not only at risk areas such as in the case of Bihar or West Bengal states of India , whole state could be targeted because of the migration of asymptomatic or partially recovered people. 

5. Table 4. Consideration for doses needed should also be given not only based on the disease burden but also for different types of vaccines. For example, the estimation of vaccine doses could vary with the type of vaccine strategies that have a long-range protection potential. 

6. While discussing the incentives for manufactures to develop vaccine for a neglected tropical disease such as Leishmania, it should be mentioned that they are other incentives by which the manufactures are eligible under a priority review voucher program of the regulatory framework. 

7. if there is a vaccine which is protective against both VL and PKDL and has broader market, what is the rationale for making a vaccine specific for PKDL which may qualify under the regulatory framework of orphan drug or biologics program?

Reviewer #3: (No Response)

PLOS authors have the option to publish the peer review history of their article (what does this mean?). If published, this will include your full peer review and any attached files.

Reviewer #1: No

Reviewer #2: No

Reviewer #3: No

Figure Files:

Data Requirements:

Reproducibility:

References

---

## [Decision Letter · Decision Letter 1]

18 Aug 2021

Dear Mr. Malvolti,

We are pleased to inform you that your manuscript 'Human leishmaniasis vaccines: use cases, target population and potential global demand' has been provisionally accepted for publication in PLOS Neglected Tropical Diseases.

Best regards,

Dhafer Laouini

Associate Editor

Eric Dumonteil

Deputy Editor

Reviewer's Responses to Questions

**Key Review Criteria Required for Acceptance?**

**Methods**

-Are the objectives of the study clearly articulated with a clear testable hypothesis stated?

-Is the study design appropriate to address the stated objectives?

-Is the population clearly described and appropriate for the hypothesis being tested?

-Is the sample size sufficient to ensure adequate power to address the hypothesis being tested?

-Were correct statistical analysis used to support conclusions?

-Are there concerns about ethical or regulatory requirements being met?

Reviewer #1: see below summary

Reviewer #2: The authors have satisfactorily responded to my comments

Reviewer #3: The author have made suggested modifications

**Results**

-Does the analysis presented match the analysis plan?

-Are the results clearly and completely presented?

-Are the figures (Tables, Images) of sufficient quality for clarity?

Reviewer #1: see below summary

Reviewer #2: The authors have satisfactorily responded to my comments

Reviewer #3: (No Response)

**Conclusions**

-Are the conclusions supported by the data presented?

-Are the limitations of analysis clearly described?

-Do the authors discuss how these data can be helpful to advance our understanding of the topic under study?

-Is public health relevance addressed?

Reviewer #1: see below summary

Reviewer #2: The authors have satisfactorily responded to my comments

Reviewer #3: (No Response)

**Editorial and Data Presentation Modifications?**

Reviewer #1: see below summary

Reviewer #2: None

Reviewer #3: (No Response)

**Summary and General Comments**

Reviewer #1: I was the first reviewer on the original version. The authors have effectively addressed my comments and suggestions.

This is a timely and important publication and is acceptable in it's current version

Reviewer #2: The authors have satisfactorily responded to my comments

Reviewer #3: (No Response)

PLOS authors have the option to publish the peer review history of their article (what does this mean?). If published, this will include your full peer review and any attached files.

Reviewer #1: No

Reviewer #2: **Yes: **Hira Nakhasi

Reviewer #3: **Yes: **Prof. Sarman Singh, FRCP

---

## [Editor Report · Acceptance letter]

16 Sep 2021

Dear Mr. Malvolti,

We are delighted to inform you that your manuscript, "Human leishmaniasis vaccines: use cases, target population and potential global demand," has been formally accepted for publication in PLOS Neglected Tropical Diseases.

Best regards,

Shaden Kamhawi

co-Editor-in-Chief

Paul Brindley

co-Editor-in-Chief
